# New Insights on Low-Temperature Fermentation for Food

**Chen Liang [1,2], Ling-Xiao Liu [3], Jun Liu [1], Aihemaitijiang Aihaiti [1], Xiao-Juan Tang [2,*] and Yun-Guo Liu [2,*]**

[1] College of Life Science and Technology, Xinjiang University, Urumqi 830046, China
[2] College of Life Sciences, Linyi University, Linyi 276000, China
[3] Linyi Academy of Agricultural Sciences, Linyi 276012, China
[*] Correspondence: tangxiaojuan@lyu.edu.cn (X.-J.T.); yguoliu@163.com (Y.-G.L.)

**Abstract:** Fermentation technology has a long history and low-temperature fermentation has now become the focus of research. This paper reviews the mechanism and application of low-temperature fermentation and the optimization of relevant strains. Low-temperature fermentation leads to a differential expression of growth in metabolism genes (PSD1, OPI3, ERG3, LCB3 and NTH1). Low-temperature fermentation can be applied to foods and has various advantages, such as increasing changes in volatile flavor compounds and other corresponding metabolic substances of the strain, and inhibited growth of spurious bacteria. The focus of low-temperature fermentation in the long run lies in strain optimization, which is to protect and optimize the strains through a variety of methods. Low-temperature fermentation can greatly improve product quality. At present, the most effective methods to promote low-temperature fermentation are gene knockout and probiotic microencapsulation.

**Keywords:** low-temperature fermentation; fermentation mechanism; strain optimization; immobilized cells; probiotic microcapsule

## 1. Introduction

Fermented foods date back to 13,000 BC [1,2], and have at least 5000 years of history in China [3,4]. As a preservation method, fermentation has developed in tandem with the global economy [5]. It has gradually evolved from natural fermentation to single-cell fermentation with known strains, and then to mixed-cell fermentation. Now, it has become more and more standardized and industrialized with a greater emphasis on taste quality. Compared with traditional techniques, modern fermentation technologies have experienced profound changes, both in products and processes (Figure 1). Fermentation is a process in which large organic molecules are broken down into simpler molecules through the action of microorganisms [6]. For example, protein, carbohydrate and fat in the substrate are broken into amino acids, organic acids, peptides, alcohols, esters and other substances under the action of microorganisms. Through this action, new substances are generated to improve the flavor and taste of the original product and increase its antioxidant and anti-inflammatory functions [1,7]. There are many kinds of fermented foods, which can be divided into plant-based fermented foods and animal-based fermented foods according to the raw materials used. Plant-based fermented foods include grains, fruits, vegetables and tea. Animal-based fermented foods mainly include milk, meat and eggs (Table 1). According to the product type, fermented foods can also be divided into wine, milk, vinegar and flour products [8].

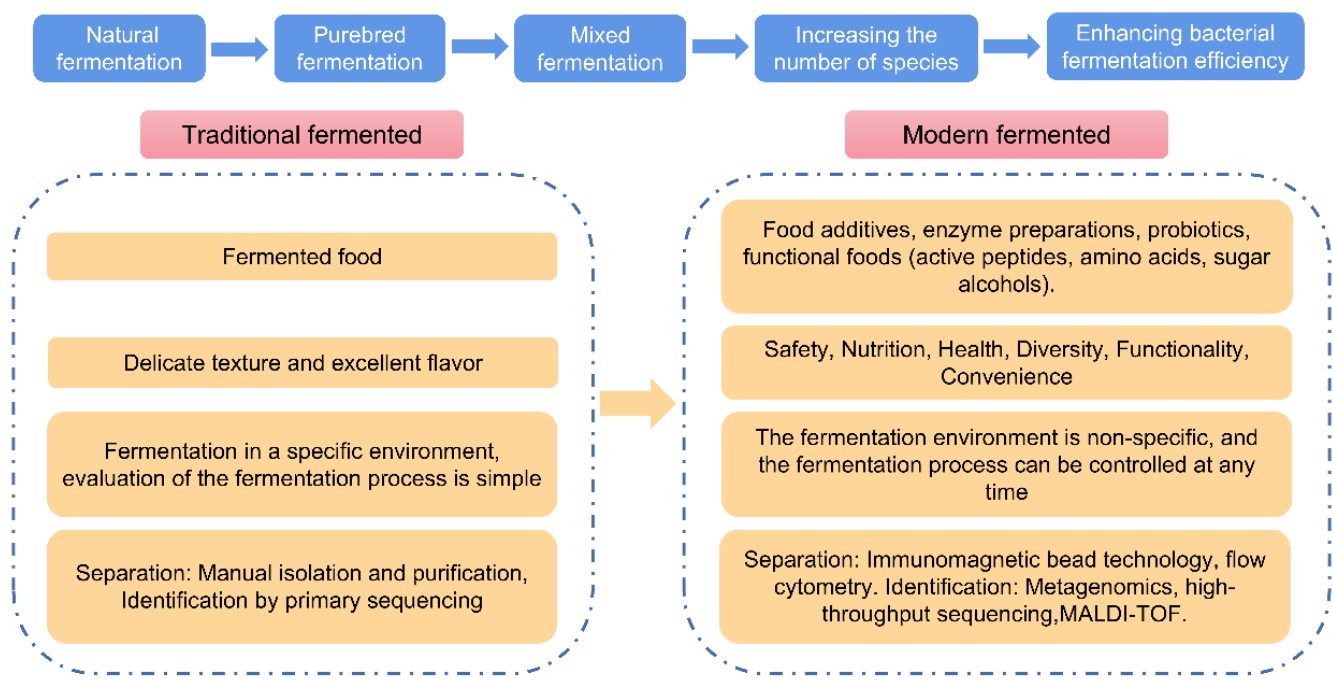

**Figure 1.** Evolution and difference of fermentation methods.

**Table 1.** Classification of fermented food and some fermented products.

| Classification | Fermented Product | Fermentation Condition | Strain | Place of Origin | Reference |
|---|---|---|---|---|---|
| **Fermented food of animal origin** | | | | | |
| **Egg** | Egg yolk | 42 °C | *Streptococcus thermophilus, Lactobacillus delbrueckii* ssp. *Bulgaricus* | CHN | [9] |
| **Livestock and Poultry** | Jerky | 35 °C | *Lactobacillus sakei, Lactobacillus plantarum* | CHN | [10] |
| | Sausage | 20 °C | *Lactiplantibacillus plantarum* | AR | [11] |
| | Ham | 30 °C, 16 °C, 12 °C | *Lactobacillus plantarum, Staphylococcus xylosus* | CHN | [12] |
| **Seafood** | Shrimp sauce | natural fermentation | *Flavobacterium, Tetragenococcus* | CHN | [13] |
| | Fish | 20 ± 5 °C | *Fusobacterium, Psychrilyobacter, Psychromonas* | CHN | [14] |
| **Dairy** | Cheese | 43 °C | *Streptococcus salivarius subsp. Thermophilus, Lactobacillus delbrueckii subsp. bulgaricus* | BRA | [15] |
| | Koumiss | 22–26 °C | *Lactobacillus* | CHN | [16] |
| | Yogurt | 30 °C | *Streptococcus thermophilus, Lactobacillus delbrueckii subsp. bulgaricus* | IN | [17] |
| | Kefir | 25 °C | *Lactobacillus, Lentilactobacillus, Leuconostoc* | AUS | [18] |
| **Fermented food of plant origin** | | | | | |
| **Fruit** | Beverage | 37 °C | *L. paracasei* | THA | [19] |
| | Vinegar | 25 °C | *Acetobacter Lacticaseibacillus Leuconostoc sp* | **TUR** | [20,21] |
| | Juice | 37 °C | *Lactiplantibacillus plantarum* | BRA | [22] |
| | Wine | 15 °C | *S. cerevisiae* | AR | [23] |
| **Vegetable** | Chili paste | 30 °C | *Lactiplantibacillus, Lactobacillus, Weissella, Issatchenkia, Trichoderma, Pichia* | CHN | [24] |
| | Pickles | 25 °C | *L. fermentum, L. plantarum* | CHN | [25] |

Table 1. *Cont.*

| Classification | Fermented Product | Fermentation Condition | Strain | Place of Origin | Reference |
|---|---|---|---|---|---|
| **Grain** | Natto | 40 °C | *Bacillus subtilis* | JAP | [26] |
| | Bean paste | 20 °C | *Aspergillus oryzae* | CHN | [27] |
| | Bean curd | natural fermentation | *Bacillus, Enterobacter, Lactobacillus, Sphingobacterium, Stenotrophomonas, Tetragenococcus, Trabulsiella, Weissella Alternaria, Sterigmatomyces, Actinomucor, Fusarium, Debaryomyces, Candida* | CHN | [28] |
| | Soy sauce | 37 °C | *A. oryzae, Z. rouxii, L. fermentum, K. kristinae* | CHN | [29] |
| | Rice cake | 32 °C | *Lactobacillus plantarum, Saccharomyces cerevisiae, Candida humilis* | CHN | [30] |
| | Plant-based meat analogue | 20 °C, 25 °C, 30 °C, 35 °C | *Rhodotorula mucilaginosa, Monascus purpureus* | CHN | [31] |
| | Bread | room temperature | yeast, *Lactiplantibacillus plantarum* | ES | [32] |
| | Steamed bread | 35 °C | *Saccharomyces cerevisiae*, Baijiu Qu | CHN | [33] |
| | cereal vinegar | - | *Lactobacillus, Acetobacter* | CHN | [34] |
| | Rice wine | 25 °C→32 °C→20 °C | *Pediococcus, Bacillus, Monascus, Saccharomyces, Rhizopus* | CHN | [35] |
| | Beer | - | *Saccharomyces cerevisiae, Saccharomyces pastorianus* | BRA | [36] |
| | Chinese baijiu | 30 °C | *Lactobacillus, Aspergillus* | CHN | [37] |
| **Other** | Tea | 30 °C | *Komagataeibacter oboediens* | CHN | [38] |

The basis of fermentation is microorganisms, and temperature is an important factor affecting the activity of microorganisms. Therefore, optimal temperature is required to ensure and accelerate the fermentation process when conditions permit. Low-temperature fermentation means that the fermentation process runs smoothly with reduced temperature. No fermentation, fermentation, low-temperature fermentation—these three methods have their own advantages and disadvantages (Table 2). Low-temperature fermentation can improve the appearance, shape and shelf life of products, and prevent diseases such as hypertension, diabetes, hyperlipidemia and obesity [39], which are similar to the role of ordinary fermentation. In addition, low-temperature fermentation has some unique advantages. Practical use shows that low-temperature fermentation can stay under stable and reliable control. At low temperatures, the growth and reproduction of microorganisms slow down, and the fermentation process becomes slower, smoother and more complete. In terms of changes in metabolic components, it forms small molecules that are not easily broken down and optimizes metabolic products, resulting in complete conversion of sugars and changes in the content and proportion of acids, alcohols and volatile flavor compounds that provide aroma. In some fruit wines, the wine body will be clearer, the taste will be better, and the unique flavor of the raw materials will be preserved [40]. In terms of nutritional function, it fully extracts various functional factors and nutrients from raw materials to produce prebiotics [41,42]. The release of these bioactive substances improves anti-inflammatory, antioxidant and other activities. In terms of strains, it improves the number of live bacteria and also leads to some antibacterial activity [43,44]. For example, human pathogenic bacteria such as *Bacillus cereus* grow more violently in normal temperature fermentation than in low-temperature fermentation, and the pollution of products is more prominent [45]. The addition of low-temperature tolerant lactic acid bacteria to silage ensures the smooth fermentation process at low temperature, and the combined effect of acid and low temperature inhibits the fermentation of undesirable Clostridium [46]. Low-temperature two-stage fermentation (20–10 °C) for bream effectively inhibits the growth of Pseudomonas and Coliforms [47]. The low temperature fermentation of Longyan wine at 10 °C effectively inhibits wine spoilage and improves the quality of the wine [40].

However, there are also some disadvantages. For example, microbial decarboxylation of amino acids will generate biogenic amines, which are harmful to the human body and prevalent in fermented fruits and vegetables [48]. Tyramine tends to stay in meat [49], and cereal products such as alcohol contain urethane, which is to some extent carcinogenic [50]. However, the synthesis of biogenic amines such as cadaverine and putrescine is inhibited in low-temperature fermentation [47], indicating that low-temperature fermentation can improve product quality and safety through the inhibition of undesirable strains of bacteria and reduction of undesirable toxins. Many years ago, China used this long-time low temperature fermentation technology to ferment pickles and soybeans and store them in a sealed and cool place for a long time [51]. In recent decades, research on low-temperature fermentation become extensive and intensive. It was initially applied to pickles and fruit wine, and then to fermented foods such as dairy and flour products. It can be seen that low-temperature fermentation is now leading a new trend of fermented products.

**Table 2.** Advantages and disadvantages of different fermentation methods.

|  | No Fermentation | Fermentation | Low-Temperature Fermentation |
|---|---|---|---|
| Advantages | 1. Short production cycle; flexible timing 2. Better retention of the original flavor and the texture of the ingredients. | 1. Moderate length of time between non-fermentation and low-temperature fermentation. 2. Good effect of reducing sugar. 3. Better mouthfeel and taste than unfermented products. 4. Effectively extend the shelf life. | 1. Effectively reduce microbial contamination in the environment and inhibit the growth of stray bacteria. 2. Slow down the reaction process, resulting in more complete metabolism and an increase in the variety and content of flavor substances. 3. Effectively extend the shelf life. |
| Disadvantages | 1. Poor taste and flavor. | 1. Relatively long production cycle and complex production process. 2. Fast and uncontrollable fermentation process. 3. Susceptible to miscellaneous bacteria. | 1. Longer production cycle and complex production process. 2. Fermentation failure may occur. |

## 2. Mechanism of Low-Temperature Fermentation

The mechanism action of low temperature fermentation is shown in Figure 2. From a macro perspective, low temperature reduces the activity of microorganisms, decreases the fermentation and metabolic rate, delays the completion of fermentation and slows down the enzymatic reaction, thus affecting the synthesis of its metabolites. It increases the fermentation time and makes the metabolic reactions complete. Low temperature will change the structure of the original flora and the metabolic pathway by affecting the activity of fermentation strains. Different strains have different regulatory mechanisms on the yield of compounds [52]. All of these changes have an impact on the final flavor.

From a microscopic perspective, the core causes are differential gene expression and structural changes [53]. The metabolic fluxes are basically the same but the concentration of growth-limiting nutrients, such as glucose and ammonia, vary greatly [54]. The differential expression of genes leads to the emergence of multiple changes in protein transcription translation folding, amino acid biosynthesis, lipid metabolism, membrane fluidity, mitochondrial function, mRNA secondary structure stability and metabolic enzyme activity. This leads to changes in aldehyde metabolism, lipid metabolism, polysaccharide metabolism, energy metabolism, alcohol metabolism and many other pathways [54,55]. Research has shown that yeast is far more adaptable to low temperatures than lactic acid bacteria and acetic acid bacteria. Therefore, the most popular application of low-temperature fermentation is fermented wine. In the case of *Saccharomyces cerevisiae*, in the initial stage of fermentation, the transcriptional process expresses the cold stress response. Cold shock specifically induces a set of proteins that are involved in transcription, translation and other essential functions that help maintain the structure of nucleic acids [56,57]. Up-regulated

genes related to the cell cycle, growth control and maintenance of exponential growth phase are transcribed earlier than in normal fermentation [58].

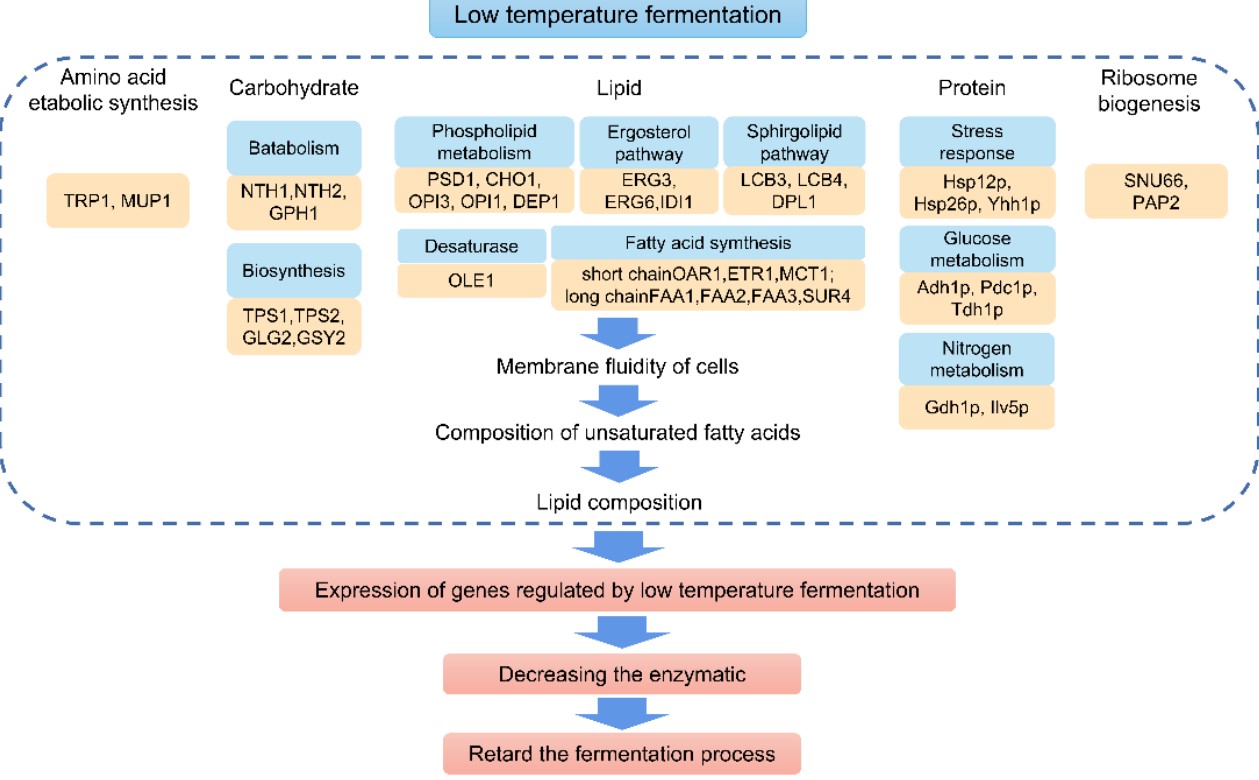

**Figure 2.** The mechanism of low-temperature fermentation.

Differences caused by the expression of genes that regulate lipid metabolism alter the composition of lipids. Overexpression of lipid genes (PSD1, LCB3, DPL1 and OLE1) improves fermentation activity during low temperature fermentation, and overexpression of the OLE1 gene related to unsaturated fatty acids produces a specific aroma profile in wine [59]. The fermentation temperature will affect lipid unsaturation and composition, and the low fermentation temperature will increase the content of medium chain fatty acids [60]. Genes such as FAA3, FAA1 and SUR4 are associated with cytoplasmic fatty acid synthesis and encode enzymes required for mitochondrial short-chain fatty acid synthesizing. Changes in these genes result in reduced phospholipid content and membrane fluidity: the content of short- and medium-chain fatty acids increased; the content of long-chain fatty acids decreased; and the content of esters increased. At the same time, an increase in resistance to ethanol was found in low-temperature fermented wines [61].

The transcription and translation of proteins and the biosynthesis of amino acids are the key factors affecting flavor. Amino acids are prerequisites for flavor and are closely associated with the change of flavor. At low temperatures, nine proteins involved in stress response, gluconeogenesis and nitrogen metabolism were significantly altered, and the overexpression of ILV5 gene increased the low-temperature fermentation activity in a short period of time [62]. There are eight unique molecular functional proteins fermented at 13 °C in *Saccharomyces bayanus var uvarum*. Proteins up-regulated at 13 °C were associated with temperature stress and the production of aromatic compounds involved in amino acid metabolism, as well as the production of heterocyclic alcohols and their derivatives [63]. Proteomic differences were analyzed in three low-temperature tolerant yeasts, *Saccharomyces cerevisiae*, *Saccharomyces uvarum* and *Saccharomyces kudriavzevii*, which have increased available ribosomes at low temperatures and overexpression of SNU66 and PAP2, genes related to ribosome biosynthesis [64]. With the decrease in fermentation temperature, the ability and efficiency of protein translation, amino acid metabolism and

biomass production increase [58]. The genome-wide analysis of *Saccharomyces cerevisiae* under low temperatures reveals a compensation mechanism under low temperature stress, which increases the ability and efficiency of translation, as well as the number of proteins involved in translation, and up-regulates the genes involved in translation. Among them, proteins with cold-sensitive genes involved in amino acid biosynthesis are significantly enriched, especially aromatic amino acids. Translation efficiency is a key limiting step in low-temperature fermentation, which is consistent with the conclusion of both.

Tryptophan is the limiting amino acid of *Saccharomyces cerevisiae* at low temperatures. During the low-temperature fermentation of wine, deletion of TRP1, a gene that controls tryptophan metabolism, decreases the rate of amino acid metabolism, but does not affect the rate of fermentation. On the contrary, the deletion of TRP2 increases both the amount of nitrogen consumed and the fermentation activity, while decreasing the consumption of ammonium [65]. Low temperatures reduce nitrogen consumption, partly because of the low expression of genes controlling ammonium and amino acid permeases, and partly because of reduced membrane fluidity, which affects osmotically active osmotic activity, all of which affect nitrogen catabolism, amino acid transport and nitrogen uptake [66]. Although the quality of low temperature fermented wine will be improved, additional nitrogen sources need to be added to reach a state similar to dynamic equilibrium to better meet the needs of fermentation. *Saccharomyces cerevisiae* and some non-*Saccharomyces cerevisiae*, especially *Torulaspora delbrueckii*, have similar nitrogen source consumption paths in fermented wine [67].

Another equally critical pathway is the sulfur assimilation pathway. MUP1 is associated with the uptake of methionine and cysteine, and URM1 is associated with sulfur assimilation, both of which significantly affect low-temperature fermentation [68]. Both the sulfur assimilation pathway and glutathione biosynthesis genes are upregulated to better activate the synthesis of key metabolites, such as glutathione, which has a certain protective effect on the fermentation process [69]. In the later stage, low-temperature fermentation can be protected and optimized by supplementing glutathione, which is confirmed by the production of glutathione from low-temperature rice [70]. The same author analyzed the three metabolic pathways of *Saccharomyces cerevisiae* and found that the subunit encoded by the GAA1 gene could add the GPI required for inosine synthesis to newly synthesized mannose proteins, and that inosine limitation altered the efficiency of low-temperature fermentation [71].

## 3. Application of Low-Temperature Fermentation in Food

### 3.1. Grain Products

Xu et al., found that in the process of making bread using mixed leavening agents (*Meyerozyma. guillermondii*, *Pichia. kudriavzevii*, and *Lactobacillus. sanfranciscensis*), more key volatile compounds and aromatic compounds, such as alcohols, aldehydes and esters, were formed under the fermentation temperature of 10 °C than 30 °C, which improves the sensory quality of bread. At the same time, the low temperature facilitates the production of extracellular polysaccharides by lactic acid bacteria and optimizes their textural properties [72]. It provides a technical basis for the comparison of low-temperature and normal-temperature fermentation in wheat flour products. The accumulation of benzaldehyde and nonanal in bread fermented at 20 °C increases, and the extension of fermentation time increases the accumulation of alcohol and ester. Acetaldehyde also accumulates with fermentation under low temperature conditions [73]. All these can provide excellent taste and flavor to the product. The oats silage is stored with low-temperature resistant lactic acid bacteria, such as *Lactobacillus plantarum*, *Lactobacillus brucelli* and *Streptococcus pentosus*, which not only improves the contents of water-soluble carbohydrates, crude protein, fiber and other components of inoculated silage, but also inhibits the growth of undesirable *Clostridium* and better retains the nutrition of the silage [46]. Compared with commercial beer yeast, *Saccharomyces eubayanus* low-temperature fermented wort produces more

volatile compounds such as alcohols, esters and phenols, and creates some fruity and floral fragrance that commercial yeast does not have [52].

### 3.2. Dairy Products

A cooling step was added to the fermentation process and the fermentation continued at 40 °C after the temperature was reduced by 4 °C from 40 °C for a period of time. The fat-free yoghurt product improved whey retention ability and dehydration resistance [74]. The yoghurt fermented at 22 °C, after the mixing of *Lactobacillus acidophilus*, *Lactobacillus bifidus*, *Streptococcus thermophilus* and *Lactobacillus delbrueckii*. Although the overall anti-inflammatory effect of the low-temperature yoghurt against colitis was the same as that of the general fermentation temperature, the antioxidant activity was much higher than that of the normal temperature [75].

### 3.3. Meat Products

It was very challenging to ensure the safety of fermented foods with good flavor under low salt content and non-acidic conditions. Tian et al. found that two-stage fermentation (20–10 °C) could reduce the accumulation of biogenic amines and slightly promote the formation of flavor, which could be an alternative to the production of high-quality low-salt fermented bream [47]. When inoculated with *Lactobacillus plantarum* and *Staphylococcus Fahrenheit* at 30 °C, the content of free amino acids increased significantly and the content of titratable acid, organic acid, protein and fat decreased [76].

### 3.4. Fruit and Vegetable Products

The fermentation temperature modified the sensory profile of the wines. A fresh fermented persimmon beverage was obtained by adding *Saccharomyces cerevisiae* to the persimmon pulp and fermenting at 25 °C for 10 days [77]. Massera et al. studied the effect of fermentation temperature and yeast strain on the aroma profile of Merlot wines elaborated at a pilot scale. They proved that low-temperature fermentation produced wines with higher ester and lower terpene content than at 25 °C [23]. When wine is fermented at low temperatures, the improved taste and flavor of the products is because that the fruit aroma increases, and the aroma of each part is better balanced [78]. Shi et al. found that, compared with the two temperatures of 15 °C and 20 °C, the content of esters that express mint and fruit flavor increased after the fermentation of Longyan wine at 10 °C, and the aroma intensity and overall balance improved [40]. The amount of aromatic compounds increased as a result of changes in some unique microorganisms under low-temperature conditions. The low-temperature-tolerant strain *Saccharomyces uvarum* was isolated from low-temperature natural fermentation cider to ferment cider, and made ethanol and sulfite tolerant [79]. *Weissella koreensis* is the dominant microorganism of fermented kimchi at 0 °C for 2–3 months. The strain has a low degree of carbohydrate assimilation and a strong ability to produce ornithine. It grows well under mild and cold conditions [80]. The volatile components and the pungent odor of kimchi fermented at low temperature are different [81].

In addition to low-temperature fermented food, low-temperature fermentation can also be used to produce fat, ethanol and other substances. The yield of α-linoleic acid produced by Oleaginous yeast *Yarrowia lipolytica* by low-temperature cultivation was improved up to 3.2 times of standard growth conditions [82]. Compared with fossil fuels, biofuel ethanol prepared by sugarcane has a better effect on environmental protection. Fermenting sugarcane with vacuum-speed steaming and low-temperature yeast fermentation can increase the ethanol yield and decrease the remaining amount of lees [83,84].

## 4. Low-Temperature Fermentation Strain Optimization Method

To achieve sustainable development of low-temperature fermentation, the selection and optimization of strains is essential. At present, the main methods available for reference and implementation are screening, domestication, cell immobilization, microencapsulation

and the addition of some strain protectants (Figure 3). There are also other advanced methods, such as genetic optimization.

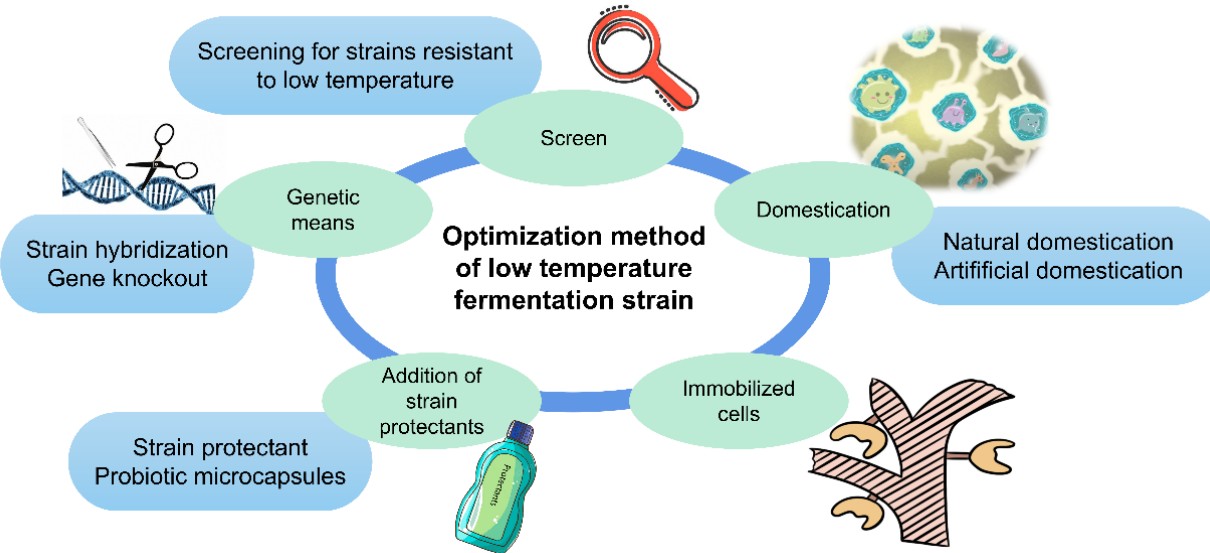

**Figure 3.** Optimization methods for low-temperature fermentation strain.

### 4.1. Screen

For the screening of unknown strains, the desired strains can be obtained by isolating, purifying and identifying strains from existing natural fermentation samples, and inoculating the obtained strains back into the samples. The application of the strains to low-temperature fermentation can standardize the fermentation process and prevent fermentation failure caused by low temperature. Screening from fermented products can obtain strains that are more suitable for the characteristics of the raw materials. Lactic acid bacteria (LAB) isolated from blueberries are used in fermented blueberry juice. The strains are more suitable for fermented blueberry juice [85]. Gunduz isolated the lactic acid bacteria from the collected sourdough samples and obtained two kinds of lactic acid bacteria, *Fructilactobacillus sanfranciscensis* and *Lactiplantibacillus plantarum*, which were inoculated into the dough and fermented at 28 °C. The fermented sourdough showed a significant increase in the acidification rate and the profile of volatile organic compounds (VOC) [86]. Lancetti and Yeşi et al. used different *Limosilactobacillus fermentum*, isolated from quinoa and buckwheat, to ferment rice, buckwheat and quinoa flour mixed dough at 30 °C. The flour properties changed significantly and the antioxidant activity improved [87,88]. He isolated *Lactobacillus plantarum*, *Candida dwarfism* and *Saccharomyces cerevisiae* from the rice syrup of fermented rice cake [30]. The quality of the rice cake fermented at 32 °C for 8 h was, to some extent, better than that of commercially available products. Under the premise that the microbiological indexes were standard-compliant and the physicochemical indexes did not differ much, the volatile components were richer, more concentrated and sensorily better. The quantity and total amount of volatile flavor substances increased [30]. Bachtarzi screened extracellular polysaccharides producing LAB from dairy products for skim milk fermentation, which can be used as stabilizers and thickeners and are safer and more nutritious than additives [89]. Li screened the dominant strains *Lactobacillus fermentum* and *Candida santamariae* from the natural fermentation broth produced by rice flour, adjusted the proportion, and then applied the rice flour fermented back to rice flour to improve the hardness, chewiness and taste, color, fragrance and taste of rice flour. That is to screen foods that have been fermented at low temperature and optimize the strain ratios to obtain low-temperature fermentation products [90].

Lactic acid bacteria with strong proteolytic ability can improve flavor and taste after fermentation because they can degrade large molecules of proteins into small molecules of peptides and form flavor substances. Fermented mung bean milk showed higher ACE

inhibitory activity and proteolytic percentage at the end of fermentation, so *Lacticaseibacillus casei* and *Lactococcus lactis* were selected to ferment mung bean powder [91]. Delgado and Hwang fermented soybean milk with *Lactobacillus plantarum* and *Lactobacillus brevis*. The total average flavor decreased, and enzyme inhibitory activity and antioxidant capacity reached the peak [92,93]. Ramos passed a pair of γ-aminobutyric acid production to screen the pre-existing lactobacilli. *L. paracasei* and *L. plantarum* exhibited high concentrations in the fermentation of amniotic milk γ-aminobutyric acid [94]. After fermenting ginseng sprouts with *Lactobacillus plantarum* at 30 °C for three days, it was found that the antioxidant capacity increased, indicating that it is more suitable for functional medicine than unprocessed ginseng sprouts [95]. This provides inspiration for the application of low-temperature fermentation. One feasible way is to select available strains that are suitable for low-temperature. This was achieved by lowering the incubation temperature, trying different strains of bacteria and exploring the growth features of the strains, and selecting strains according to the raw materials and needs of fermented products. It is also possible to select directly from strains that have been applied to other low-temperature fermentation products and apply them directly to the target products.

### 4.2. Domestication

Domestication can be divided into two types. One is natural domestication. In fact, extreme environments can enable natural domestication. Strains that are more adaptable to the environment can be obtained by screening strains from the environment. For example, high-temperature resistant yeast strains are screened in sugarcane fields in Thailand [96]. Then, fermentable strains can be screened in a low-temperature environment and applied to low-temperature fermentation. The other is artificial domestication, which is one of the commonly used methods in the laboratory. This method is to connect the fermentation strains to the corresponding liquid culture medium and place them in the temperature gradient for cultivation in turn. It can gradually adapt to the low temperature environment and finally obtain the desired strains at the required temperature after low-temperature domestication as a seed source for subsequent fermentation.

### 4.3. Immobilized Cells

At present, immobilized cells have been applied in many fields, such as food, environment and energy. The key technology of immobilized cells lies in the ability to immobilize strains on different materials in order to add or enhance one or more characteristics. Gluten granules and other structurally stable materials bear the brunt. As one of the immobilized materials, *Saccharomyces cerevisiae* strains are immobilized on gluten granules and then fermented in wort at low temperature. The beer obtained not only has an alcohol content within the range, but also boasts excellent clarity, aroma and taste [97]. A new biocatalyst was prepared by immobilizing it on whole wheat grains, which can not only complete high quality fermentation in an extremely low temperature environment of 5 °C to obtain clear wine, but also maintain stability for a long time. Thirty cycles of fermentation can also be completed. After the culture has expanded, fermentation at 20 °C and 2 °C does not affect its fermentation ability. Compared with free cells, the wine fermented by immobilized cells increases the formation of esters. Aromatic characteristics improved, while the percentage of total esters increased and the percentage of higher alcohols decreased at lower fermentation temperatures [98]. Immobilizing this strain on dried fig not only brings about a better fermentation effect, but also retains the special aroma and taste of the figure. With the decrease of fermentation temperature, the concentration of ethyl acetate increases and the concentration of pentanol decreases. The beer produced by cells immobilized on gluten particles or other food-grade carriers, such as lignocellulose materials, is more concentrated [99]. Therefore, in low-temperature fermentation, we can fix the desired strain on gluten particles, cellulose and other fixed materials for experiments, which can add advantages while fermenting.

Another category of materials that is widely studied at present is cellulose. Peanut shells and coconut shells belong to agricultural waste with a high proportion. Their main components are lignin, cellulose and hemicellulose. Tubular cellulose is used as a fixture for *Saccharomyces cerevisiae* and *Lactobacillus casei*. After fixation by thermal drying, it is used in beer and dairy fermentation. The content of fruity esters in fermented milk increased, the dimethyl sulfite and o-dione decreased, and the ratio of ester to alcohol increased in beer. The products fermented at 18 °C had higher concentrations of acid substances and total volatile flavor substances, and a lower effect on alcohols and esters [100]. Immobilizing the low temperature and alcohol resistant *Saccharomyces cerevisiae* strain in porous cellulose to ferment maltose at an extremely low temperature of 5–10 °C can greatly accelerate the rate of maltose fermentation. The presence of porous cellulose increases the uptake of maltose by immobilized cells, which is due to the hydrogen bond attraction on the surface of porous cellulose and the continuous pumping of maltose to cells [101]. When the temperature is continuously reduced from 30 °C to 10 °C, the alcohol content obtained after fermentation will continue to increase. This sheds new light on domestic fermented wine; that is, after adding water to dry materials and putting them into the refrigerator for low-temperature fermentation, manual fermented cherry wine can be obtained. The ingredients are clean and safe. At the same time, immobilized yeast can not only solve the problems of slow fermentation and low fermentation efficiency, but also reduce microbial pollution. The fatty acid ethyl ester and higher alcohol acetate formed will also produce flower and fruit fragrances [102]. The delignified cellulose biocatalyst freeze-dried on delignified cellulose is suitable for low-temperature winemaking (5–15 °C) and produces wines with 12% alcohol content [103].

### 4.4. Add a Certain Strain Protection Agent

#### 4.4.1. Probiotic Microcapsules

Microencapsulation is currently the most effective and promising method for strain protection. Encapsulated substrates provide effective protection for microorganisms from environmental extremes and other external factors. In the case of dietary supplements, for example, probiotics must withstand adverse environmental conditions and reach the target site in sufficient numbers, mainly due to the regulation of the human intestinal microbiota. Embedding strains with bioactive substances not only improves tolerance to the environment, but also adds effects that were not previously present, such as antioxidant activity [104]. Wang et al. embedded *Bifidobacterium adolescentic* in 10.00% (*w/w*) chickpea protein isolate and 0.20% (*w/v*) genipin, 0.20% (*w/v*) alginate κ- Carrageenan crosslinked wall materials. It can slowly release bacteria in microcapsules under the extreme environment of pH 2.0 and temperature of 25 °C, which ensures a smooth fermentation process under extreme conditions [105]. Adding the alginate of *Robinia pseudoacacia* bean to entrap *Lactobacillus rhamnosus* microcapsules can better delay large-scale release [106]. Pandey and Mishra developed an encapsulated powder of γ-aminobutyric acid and *Lactobacillus plantarum* using spray drying in a biocompatible matrix composed of inulin, dextran, and maltodextrin [107]. After 120 days of storage at 4 °C, there was no difference in bacterial viability as compared with the fresh microcapsules. In addition, some plant extracts [108,109], gums [110,111] and dietary fibers [112] are used in the production of microencapsulated wall materials to provide different effects. This provides inspirations for the application of low-temperature fermentation. It can embed low-temperature fermentation bacteria in suitable wall materials to prevent the strain from being immediately and completely exposed to the low-temperature environment, reduce adverse effects on microorganisms, delay the release of the desired strain, and ensure viability and duration of action of the strain during low-temperature fermentation. Fermentation bacteria can be prepared into microcapsules before use, which can greatly improve the adaptability of bacteria to low temperature and the vitality of fermentation. Theoretically, any wall material suitable for low temperatures can be used for all low-temperature fermentation strains.

### 4.4.2. Strain Protectant

The three protective mechanisms of strain protectants prevent the formation of intracellular and extracellular ice, the water displacement hypothesis formed by hydrogen bonding, and the formation of the glassy matrix [113]. At present, studies have shown that some sucrose, maltose, trehalose, maltodextrin, skim milk powder and other disaccharides, alcohols, proteins and antioxidants can be added as strain protectants at low temperatures to maintain the survival rate of strains and the stability of cells. The addition of trehalose as a cryoprotectant improved the survival rate of probiotics in freeze-dried calcium alginate coated microcapsules from 3% to 41.26% [114]. The addition of lactose alcohol had a positive effect on the survival of encapsulated bacteria. It also has a certain effect when at cold storage and freezing temperature, indicating that lactose alcohol can also be an effective protective agent [115]. The new protective medium containing sucrose (10%), trehalose (10%), skim milk (10%) and antioxidants also has a certain protective effect on bacteria under different storage conditions [113]. In the process of low-temperature fermentation, some sugar substances and proteins can be added as strain protectors to smoothen the fermentation process.

### 4.5. Genetic Means

The hybridization of strains is to select the performance characteristics required in the parents for hybridization. The obtained hybrid strains can achieve the comprehensive characteristics of the two parent strains. Although the operation of strain hybridization is relatively simple, the production cycle is lengthy. Hybrids were constructed using three strains of *Saccharomyces. eubayanus*, *Saccharomyces. uvarum* and *Saccharomyces. Cerevisiae*, selected to be resistant to low nitrogen and low temperature. It was found that a better fermentation rate was achieved under low temperature or low nitrogen conditions. The hybrid strains also produced a large amount of acetate and a higher content of alcohols, which increased the intensity and complexity of the wine aroma. The hybrid strain consumes the nitrogen source faster, making it more competitive under nitrogen deficiency conditions [116]. The hybrid yeast of several new yeasts, *Saccharomyces. cerevisiae* and *Saccharomyces. kudriavzevii*, isolated from wine, performed well at low temperatures and increased the yield of alcohols [117]. Secondly, it can also be improved by means of gene editing, such as gene knockout. *Saccharomyces. cerevisiae* with strong sugar fermentation, strong acid tolerance and rapid gas production. Increased respiration can be achieved by knocking out the rgt2 and snf3 genes encoding glucose sensors in isolated strains, and also by eliminating transcription factors that encode nitrogen metabolites repression and increased aspartate consumption [118]. This gives us two new methods that can be applied in low-temperature fermentation. One is to cross the strain that can be fermented at low temperature with another strain that has the properties required to build a new hybrid, which can have the characteristics of both parents and be used in low-temperature fermentation. The second is to use gene editing and other means to eliminate genes that are stressed by low temperature and thus affect the fermentation process of the strain to achieve a smooth low temperature fermentation.

## 5. Expectation

This paper reviews the mechanism of low-temperature fermentation, its application and the optimization of low-temperature fermentation strains. The main difference of low-temperature fermentation lies in the fact that it leads to differential expression and changes in genes that affect the growth and reproduction of fermenting strains and their metabolism. Low-temperature fermentation can be applied to food with various advantages, such as increased changes in volatile flavor compounds and other corresponding metabolic substances of the strain, and inhibition of the growth of spurious bacteria. The key to the sustainable development of low-temperature fermentation lies in the optimization of strains, which is to protect and optimize the strains through various methods so as to

greatly improve product quality. At present, the most effective methods for promoting low-temperature fermentation are gene knockout and probiotic microencapsulation.

In summary, the product quality achieved through low-temperature fermentation is higher than that of normal-temperature fermentation. Low-temperature fermentation can improve the flavor and taste of fermented food and effectively avoid contamination. However, this depends on the adaptability of the strains. We know that few strains work at low temperatures, and they not necessarily produce desired efficacy. In the future, with improved knowledge on strain optimization, we can select low-temperature fermentation strains to achieve optimal food quality.

**Author Contributions:** C.L. prepared the complete draft of the manuscript and is the major contributor. L.-X.L., J.L. and A.A. provided the necessary information for and partially wrote the article. X.-J.T. and Y.-G.L. conceptualized, corrected and finalized the manuscript, and also supervised the work. All authors have read and agreed to the published version of the manuscript.

**Funding:** This work was supported by the Key Research and Development Program in Shandong Province (2019YYSP026), the High-level Talent Introduction Project of Linyi University (LYDX2018BS032), and the Innovation team of the introduction and education plan for young and innovative talents in Shandong Provincial University (2021QCYY007).

**Institutional Review Board Statement:** Not applicable.

**Informed Consent Statement:** Not applicable.

**Data Availability Statement:** Not applicable.

**Conflicts of Interest:** The authors declare no conflict of interest.

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
