# Peer review of "New Insights on Low-Temperature Fermentation for Food"

_fermentation, doi:10.3390/fermentation9050477_

Round 1

Reviewer 1 Report

Lines 35-38: this definition is closer to digestion, not fermentation. Revise in terms of energy production and conditions.

Lines 42-45: In a table, provide recent statistics for some fermented products worldwide.

Figure 1. Check and correct "cytometry.etc" - "sequencing.MALDI-TOF.etc."

Line 69: write in italics "Bacillus cereus"

Line 83: It is recommended to make a figure showing the mechanism of low-temperature fermentation.

Reviewer 2 Report

his paper seems to refer to insight from a low-temperature perspective rather than a review form for low-temperature fermentation. I wish I could choose the format of paper more clearly.
As in the epigenetics of low-fermented microorganisms, I think it's right that genes respond to all factors because they haven't been identified. But I don't think it's only at low temperatures. Abstract I think it needs to be modified a little.
Ref.10 is a review paper, so please find the actual paper and put in ref.no .
line 104, ref dealing with the substantive clause phenomenon without citing the review paper on cold shock protein in fermentation. Please present it.

Wouldn't acid production be a greater inhibitory effect on inhibiting food poisoning bacteria in Line 183, Theoats silage? Of course, I don't deny the effect of low-temperature fermentation, but I think it's a minor effect. Please state your opinion on this.
In addition, it is evaluated that the "Quality and safety considers of low-temperature fermentation" part did not cover many parts, although it dealt with the parts that should be dealt with overall low-temperature fermentation. Please supplement this part.

Reviewer 3 Report

Regarding MS entitled ‘’ New insights on Low-Temperature Fermentation for Food: A Review ‘’ this is an interesting review, but I have some comments ti improve it.

Figure 1 should be presented in a better way for the readers, please revise

L53. Don’t start the sentence with like

Table 1. please revise the numbering and the classification should be more clear

Figure 2 should be presented in clear view for the readers.

2. Mechanism of low-temperature fermentation, please revise this section, it is not well organized and it jumps from point to another point.

A figure for the mechanism of actions of the low temperature fermentation should be added.

L172. Add ref. number.

Again figure 3. Magnify the figure for  a better view

Reviewer 4 Report

Remarks below

How did the discovery of the microscope affect fermentation? Figure 1 requires a deeper explanation.

In the opinion of the reviewer, Latin names of microorganisms should be written in italics. Examples of fermentation products would be useful in Table 1. Not all foods require fermentation, and often fermentation processes reduce flavor.

What do the authors mean by low temperature? Does it concern, for example, a process carried out at a lower temperature by the same microorganisms? Or by new microorganisms and under new conditions, but for the same substrate and expected product?

Authors too often use the term "low temperature" to refer to lowering the temperature of the process of obtaining food by various fermentations. In the opinion of the Reviewer, the Authors should explain that they mainly describe fermentation processes where the temperature of the process has been reduced only by a few degrees. The Authors also do not specify how lowering the process temperature affects its efficiency. Usually, the process time is extended, which is not always economically viable.

Line 228-231

In the opinion of the reviewer, this is an oversimplification. The description of the ethanol fermentation process requires a deeper analysis and comparison. If the process was profitable, it would be used on a large scale.

The described examples lack data on existing products, e.g.The quality of the New Year cake fermented at 32℃ for 8 h was to some extent better than that of commercially available products.” What does better to some extent mean?

What was the purpose of the work? In the opinion of the Reviewer, the Authors did not define it precisely and the work seems chaotically written.

Line 362-373

What is this part of the manuscript about? In the opinion of the Reviewer, it is incomprehensible. Does this apply to cold storage of probiotics? Do they concern their release in unfavorable conditions - then how do they affect the process?

In the opinion of the reviewer, isolating new microorganisms from the environment that carry out fermentation processes at a lower temperature does not exhaust the features of optimizing low-temperature fermentation.

Line 425-427

How is prebiotic encapsulation supposed to affect low temperature fermentation?

 Line 429 - 435

In the opinion of the Reviewer, the Authors did not provide sufficient evidence to improve the quality of the products obtained at a lower fermentation temperature.

Round 2

Reviewer 2 Report

The authors have tried to answer in good faith many of the problems raised in the first review. Also, I deeply appreciate your efforts to improve the completeness of the paper by preparing a lot of materials. In the revised paper, it was confirmed that the quality of the paper was much improved as a result of the revision, and I think that the fermentation characteristics at low temperatures were sufficiently reviewed. In addition, it is considered an excellent paper for providing insights related to low-temperature fermentation to many researchers.

Reviewer 3 Report

I don't have any further comments.